# Microstructure and Properties of Surface-Modified Plates and Their Welded Joints

**DOI:** 10.3390/ma12182883

**Published:** 2019-09-06

**Authors:** Tai Wang, San San Ao, S. M. Manladan, Yang Chuan Cai, Zhen Luo

**Affiliations:** School of Material Science and Engineering, Tianjin University, Tianjin 300072, China

**Keywords:** K-TIG, low carbon steel, surface modification, stainless steel, polarization curve, corrosion morphology

## Abstract

The surface of Q235 low carbon steel was modified by the metal inert-gas welding (MIG) method; a 304 stainless steel surfacing layer was fabricated to improve the properties of Q235 low carbon steel. For practical industry application, keyhole tungsten inter gas (K-TIG) welding was used to weld the surface-modified plates. The microstructure, elemental distribution, micro-hardness, and corrosion resistance of the surface-modified plates and the welded joints were analyzed. The corrosion tests of welded joints and surface-modified plates were carried out with the electrochemical method and hydrochloric acid immersion method, respectively, and surface morphology after corrosion was studied. The results show that the surface-modified plates and their welded joints were defect-free. The microstructure of the surfacing layer consisted of austenite, martensite, and ferrite; and the microstructure of the weld consisted mainly of martensite. The hardness and corrosion resistance of the surfacing layer was superior to that that of low carbon steel. The micro-hardness of the weld is higher than that of the stainless steel surfacing layer and the base material. The corrosion resistance of the surfacing layer is the best, and the corrosion resistance of the welding seam is better than that of the base material.

## 1. Introduction

Low carbon steels are used in several industrial applications owing to their mechanical properties, availability, and low cost. However, shortcomings such as low hardness and poor corrosion resistance limit their scope of application [1,2]. Owing to the excellent corrosion resistance and superior mechanical properties of stainless steels, a surface layer of stainless steel was fabricated on low carbon steel, significantly improving the hardness and corrosion resistance of the low carbon steel [3,4]. Different processes have been used to fabricate the stainless steel surfacing layer: laser/TIG (Tungsten Inter Gas) welding [5,6], friction surfacing [7], hot-roll bonding [8], and submerged arc cladding [9]. However, these processes have limitations. Operation of laser/TIG welding is complex, increasing the economic costs of surfacing. Friction surfacing is a new technology for surface modification; thus, the corresponding technology is not mature, and production efficiency is relatively low. Hot-roll bonding is only suitable for batch production, and it is difficult to fabricate special surfacing structures using this method. In submerged arc cladding, it is impossible to observe surfacing behavior, and the heat input is large. In this study, MIG welding is chosen for fabrication of a stainless steel surface layer, because of its high efficiency. In addition, the parameters of the surfacing process are easy to control, and a well-formed surfacing layer can be obtained [10].

Connection technology is often used in the practical application of low carbon steel as structural parts. The same is true for the surface modified low carbon steel to meet the needs of actual production. It is necessary to weld the surface modified low carbon steel plate. However, there is no relevant research on the welding of this kind of plate. As a new welding technology, keyhole tungsten inter gas (K-TIG) has attracted the attention of researchers. K-TIG welding technology is a high-speed, one-side welding and double-side forming welding technology, based on traditional TIG welding [11]. It combines the high-quality and cleanliness of GTAW (Gas Tungsten Arc Weld) with a depth of penetration that is unmatched by conventional arc welding or plasma welding processes [12]. The K-TIG welding method realizes the relative balance between the high arc pressure formed by high current (>300 A) and the surface tension of liquid metal in a molten pool, and forms small holes to realize deep penetration welding [13]. The K-TIG process offers advantages previously achievable only with a high-cost laser, hybrid laser, or electron beam welding. This simple process makes automated, high-quality, and deep-penetration welding method accessible to any small-to-medium sized fabricator [14].

This process is suitable for welding of austenitic and duplex stainless steels, titanium and its alloys, and nickel alloys. A lot of research has been conducted on various aspects of K-TIG welding, such as tungsten gas shielding technology, arc characteristics, welding pool stability, and process efficiency. For example, Cui S L et al. [15] studied the keyhole process in K-TIG welding of 4 mm-thick 304 stainless steel. The results showed that the K-TIG welding process was more stable by using the closed-loop control system of “one pulse one open keyhole” strategy. Xie Y et al. [16] characterized and optimized the performance of K-TIG welded A1S1 430 steel joints. Feng Z Y et al. [17] used filler materials to modify the microstructure and improve the properties of K-TIG welded joints. As a promising welding method, K-TIG was adopted to weld surface modified plates in this paper.

This study consists of two main parts: (1) Fabrication of the stainless steel surface layer on low carbon steel by the MIG method, aiming to enhance the surface properties of the steel. (2) In order to cater to the practical application better, a promising welding method, K-TIG, was adopted to weld surface modified plates, and the corresponding microstructures and properties were studied.

## 2. Experimental Procedures and Materials

### 2.1. Fabrication of Surfacing Layer on Low Carbon Steel by MIG Welding 

Due to its excellent corrosion properties, austenitic stainless steel (Type 304) was selected as the surface layer material. Q235 steel was chosen as the base material because of its strength, stiffness, and low cost. The 304 stainless steel surfacing layer was fabricated on Q235 steel using MIG welding. The dimensions of the base material were 250 mm × 100 mm × 6 mm, and the diameter of the 304 stainless steel wire was 1.2 mm. The chemical compositions of the Q235 steel and 304 stainless steel are shown in Table 1 [18,19]. As can be seen from Table 1, there are big differences between the chemical compositions of the two materials. The major alloying elements in 304 stainless steel are Cr and Ni (21% and 9%, respectively), which are not present in Q235 low carbon steel. The carbon content in Q235 steel is significantly higher than that of 304 austenitic stainless steel. Therefore, the chemical composition of some selected areas and the distribution of major alloying elements (Fe, Cr, C, and Ni) in the plate and welded joint were examined by scanning electron microscopy (SEM), equipped with energy dispersive spectroscopy (EDS).

Before fabricating the stainless steel surface layer, the surface of the low carbon steel was mechanically polished and then cleaned with alcohol to remove surface oxides and other contaminates. The MIG surfacing process is schematically illustrated in Figure 1. The parameters used in the process are shown in Table 2. Multi-pass welding was used, and the coverage between each adjacent pass was 50%. Oxide films were removed from the surface of the sample by two subsequent surfacing processes. The base material was clamped to minimize deformation during the surfacing process. After cooling to room temperature, the surface of the plates was milled to ensure that the thickness of stainless steel surfacing layer was 2 mm. 

### 2.2. K-TIG Weld of the Surface-Modified Plates

After the MIG surfacing process, the K-TIG welding method was used to join two pieces of the plates, as illustrated in Figure 2. Because the plates were relatively thick and no groove was formed, a specially designed K-TIG torch, equipped with a 6 mm-diameter tungsten electrode, was introduced. As shown in Figure 2, a water cooling system was used around the tungsten electrode to reduce burn loss of the electrode [20]. A constant direct-current (DC) welding power source was used in the welding process. The shielding gas was pure Ar with a flow rate of 25 L/min. The welding current was 430 A, and the welding speed was 5 mm/s.

### 2.3. Microstructure Characterization

To observe the microstructures of the surface-modified plate and its welded joint, metallographic samples were cut along the cross section of the plate and the weld seam. The samples were then polished and etched according to standard procedure [21]. The base material was etched by 4% nital alcohol solution for 8 s to reveal the interfacial microstructure. Due to high corrosion resistance of the stainless steel, the etchant used was aqua regia (the volume ratio of hydrochloric acid to nitric acid was 3:1) for 5 s. Following preliminary investigations, the welded joints were also etched using aqua regia for 5 s. All the samples were etched at room temperature. The microstructures of the cladding layers were characterized using XJP300 optical microscope and S4800 scanning electron microscope (Hitachi, Tokyo, Japan)(Manufacture, City, State abbreviation, Country). 

### 2.4. Alloying Elements Content and Distribution

EDS was used to measure the content and distribution of alloying elements in the surface modified plates and its welded joints. 

### 2.5. Mechanical Properties

An MHV2000 type digital micro-hardness tester (Dong Hua, Shanghai, China)(Manufacture, City, State abbreviation, Country) was used to measure the micro-hardness of the surface modified plates and its welded joints. The load was 1000 g and dwell time was 15 s. The distance between two adjacent points was 0.25 mm. 

### 2.6. Corrosion Experiment

Potentiodynamic polarization test and hydrochloric acid immersion were used to investigate the corrosion behavior of the surface modified plate and its welded joint. For the surface modified plate, the cross section was selected for the corrosion experiment. As for the welded joint, the upper surface, cross section, and longitudinal section were selected, respectively, to illustrate the corrosion resistance; the corresponding areas are shown in Figure 3. The surfaces of all the samples were polished before the corrosion test.

Based on previous research [22,23,24], 3.5% NaCl solution was used as the electrolyte (pH = 7) in the electrochemical corrosion test. The NaCl solution was prepared using analytical grade NaCl and deionized water. The test was conducted in an electrochemical workstation (LK3200A, Lanlike Chemical Electronic, Tianjin, China) with a modified three-electrode system. A platinum sheet (10 × 10 × 1 mm^3^) and a saturated calomel electrode (SCE) acted as counter electrode (CE) and reference electrode (RE), respectively. The sample was used as the working electrode (WE). Only the surface area of the samples was exposed to the electrolyte solution during the test. The test parameters included open circuit potential, and potentiodynamic and potentiostatic polarization. Before the polarization test, the open circuit potential was executed to determine the stability of the corrosion potential. The potentiodynamic polarization test was carried out to measure the corrosion potential (E_corr_) and corrosion current density (i_corr_) of the as-received, surface modified, and welded samples. Scan rate was 0.5 mV/s, and the potential ranged from −800 mV to +500 mV.

As for the hydrochloric acid immersion experiment, the samples of the surface modified plate and its welded joints were immersed in 0.5 mol/L hydrochloric acid solution at room temperature. After 24 h, the samples were taken out, washed with ethanol solution and dried. Then the surface morphology of the samples after corrosion by hydrochloric acid were observed by S4800 scanning electron microscope (Hitachi, Tokyo, Japan).(Manufacture, City, State abbreviation, Country).

## 3. Results and Discussion

### 3.1. Microstructures and Properties of the Surface-Modified Plates

#### 3.1.1. Microstructure of the Surface-Modified Plate

The macroscopic morphology of the surface modified plate is shown in Figure 4. After corroding with 4% nitric acid alcohol, the surfacing layer became bright and the base material became dark. The thickness of the surface layer and the base material are 2 mm and 6 mm, respectively. The microstructure of the surface modified plate is shown in Figure 4. From top to bottom, the plate can be divided into four parts: (a) stainless steel surfacing layer; (b) fusion zone between the base material and the surfacing layer; (c) phase change recrystallization zone of the base material; and (d) base material. Four different microstructure zones were observed. Figure 4a shows the microstructure of the surfacing layer; where the metallographic structure is not clearly seen through the etch of nital alcohol, the surfacing layer was etched by aqua regia for 5 s. The microstructure of 304 stainless steel (Figure 5b) consisted of fully austenitic equiaxed grains. Based on Schaeffler’s diagram (Figure 5a) [25,26], the chromium equivalent and nickel equivalent of the 304 stainless steel was 21% and 9%, respectively. According to the diagram, the room temperature microstructures of the 304 stainless steel surfacing layer consist of austenite + martensite + ferrite. Figure 4b is the fusion zone between the base material and the surfacing layer. In this region, the structure is a mixture of flake martensite and a small amount of pearlite; Figure 4c shows the structure of the phase transformation recrystallization zone, which is an incomplete crystallization area in which grain size is different and structure is not uniform (thus, the mechanical properties are not uniform). Figure 4d is the microstructure of the base material. As hypoeutectoid steel, its microstructures consist mainly of ferrite and pearlite. 

#### 3.1.2. Distribution of Alloying Elements in the Surface Modified Plate

The EDS line scan of Fe, Cr, C, and Ni was carried out from the surfacing layer (SL) to base material (BM), as shown in Figure 6. These main elements in the surfacing layer are quite different in content from the base material. Cr and Ni content was found to be high in the surfacing layer and then drops sharply at the interface. Before the surfacing process, the content of Cr and Ni in the base material was zero. Fe is the main element in Q235 low carbon steel, which increases significantly at the interface between the surfacing layer and the base material. The occurrence of Cr and Ni in the base material indicates the diffusion of alloying elements in the fusion zone and a good combination between the base material and the surfacing layer.

#### 3.1.3. Hardness Characteristics of the Surface Modified Plate

The Vickers hardness profile across the surface modified plate is shown in Figure 7. The measurement was taken from the bottom (low carbon steel) to the top (stainless steel surfacing layer), as indicated by the blue line in Figure 7. Five paths were selected on the sample, and the average value was taken as the final result. The hardness value of the Q235 steel base material was about 180 HV, which increased as the indentation path moved closer to the interface. This increase can be attributed to the recrystallization process, which resulted in the formation of finer grains. The hardness increased sharply upon crossing the interface. The high hardness of the surfacing layer is due to the presence of some martensite in the surfacing layer, as shown in Figure 4a. This hardness profile indicates that the MIG process can be used to fabricate a stainless steel surfacing layer on low carbon steel to enhance the surface’s properties, and the result of hardness distribution is consistent with the distribution of alloying elements obtained by the EDS line scan of the surface modified plate.

#### 3.1.4. Corrosion Resistance of the Surfacing Layer and Base Material

Figure 8 compares the polarization curves of the surfacing layer (SL) and the base material (BM) in 3.5%wt NaCl solution. The corrosion potential (E_corr_) and current density (i_corr_) of the samples, which were obtained from the polarization curves by Tafel extrapolation method, are shown in Table 3. Stainless steel possessed the highest values of corrosion potential and current density, indicating that it has the best corrosion resistance. Due to the high content of Cr in the surfacing layer, its corrosion resistance was found to be better than that of the base material.

Figure 9 shows the surface morphology after hydrochloric acid corrosion. Figure 9a is the macro-morphology of the surface modified plate. Figure 9b is the surface morphology of stainless steel after corrosion; it shows that slight uniform corrosion had taken place. Figure 9c is the surface corrosion morphology. There are many pits at different depths on the surface of the base material. This indicates that the surface of low carbon steel was seriously corroded. It also can be seen from the macro-morphology that the corrosion resistance of the surfacing layer is better than that of the base material.

### 3.2. Microstructures and Properties of Weld Joints

#### 3.2.1. Microstructure of the Welded Joint Obtained by the K-TIG Welding Method

Figure 10 shows the macroscopic morphology of the K-TIG welded surface modified plates. The fusion zone, heat affected zone, and base material can be divided clearly in the figure. The columnar crystals in the weld are greatly developed due to the rapid cooling rate and the large temperature gradient at the center and edge of the molten pool. The bright part is the portion containing Cr and Ni (melting of the stainless steel surfacing layer in the welding process resulted in dispersion of the elements into the welding seam), and the dark portion is the base material. Thus, the macroscopic morphology indicates that the weld composition is a mixture of stainless steel and base material. In other words, during the welding process, the 2 mm stainless steel surfacing layer on the upper part of the plate melted and was filled with the base material in the welding process. 

Figure 11 shows the microstructures of the weld. Four regions in the weld were selected for microstructural analysis; the corresponding positions are shown in Figure 10. The microstructure in the upper part of the weld (Figure 11a) consisted of acicular ferrite and lath martensite due to low carbon content and fast cooling rate. The microstructure in the lower part of the weld (Figure 11b) contained flake martensite due to the high content of carbon and other alloying elements. The presence of martensite increases the hardness of the weld. Figure 11c,d show the microstructure of the heat-affected zone of the base material. Figure 11c is the zone near the weld; it shows that a widmanstatten structure occurred in the microstructures, the appearance of which is due to the fast cooling rate in the cooling process. The appearance of a widmanstatten structure reduces the performance of the welded joint to a certain extent. Figure 11d shows the incomplete recrystallization zone of the weld; because the structure of the zone is not uniform, the performance of the zone is not similar.

#### 3.2.2. Distribution of Alloying Elements in the Weld

EDS line was carried out through the base material (BM), fusion zone (FZ), and weld zone (WZ), as shown in Figure 12. The chemical composition of the base material has quite a different content compared to that of the weld. Inter-diffusion occured between the elements in the weld and the base material due to the thermal effect of the welding process. The closer to the weld, the higher the amount of Cr and Ni content. Similarly, due to diffusion, Fe content in the welding seam increased in the area close to the base material. The FZ is a transitional region, wherein the amount of elements changed abruptly. The differing content of alloying elements lead to different properties and structures. In addition, because the weld seam is a mixture of stainless steel and low carbon steel, the performance of the weld seam is different from that of the stainless steel surfacing layer, which include microstructures, hardness, and corrosion resistance.

From Figure 11, it can be concluded that there are different martensite structures in the different zones of the weld. Therefore, the elemental contents of zone a, b, c and d (Figure 13) were scanned. As can be seen from Figure 13, the alloying element contents in the four regions are slightly different. In the region near the surfacing layer, the content of the alloying elements is higher, while near the base material, the content of the alloying elements is less. In other words, there is less C in the weld near stainless steel and more C in the weld near base material. C can easily form stable compounds with Cr, thus reducing the corrosion resistance of materials. Therefore, the corrosion resistance in the weld may not be uniform.

#### 3.2.3. Hardness Distribution of the Weld

The hardness distribution across the welded joint is shown in Figure 14. As shown in the figure, three hardness indentation paths were taken: (1) through the upper part of the weld; (2) through the middle part of the weld; and (3) through the lower part of the weld. In path 1, the average hardness of the stainless steel surfacing layer on both sides of the weld was about 440 HV. However, the average hardness of the weld area in the same path was about 480 HV. The higher hardness of the weld area is due to the martensitic microstructure, which was produced due to faster solidification speed of the molten pool in the stainless steel part of the weld. Paths 2 and 3 passed through the base material, heat affected zone, and the weld. The hardness profiles exhibited similar characteristics. On both sides, hardness decreased with increasing distance from the weld center. This can be contributed to the formation of a widmanstatten structure, making the hardness of the heat-affected zone higher than that of the base material. Although these three paths are different in structure, the hardness of the weld zone along the three paths remained basically the same, indicating that in the process of welding, the hardness in the weld was uniform.

#### 3.2.4. Corrosion Resistance of the Weld

Figure 15 shows the potentiodynamic polarization curves of both the longitudinal section and the upper surface of the weld. The corrosion potential (E_corr_) and current density (i_corr_), which were obtained from the polarization curves by Tafel extrapolation method, are shown in Table 4. The polarization results indicate that the corrosion resistance of the weld has improved significantly. As can be seen from the figure, the uniform corrosion resistance of the upper surface of weld (USW) is stronger than that of the longitudinal section of the weld (LSW). Compared with low carbon steel, the longitudinal section and upper surface of the weld have better corrosion resistance. However, compared with the surfacing layer, the corrosion resistance of the longitudinal section and the upper surface of the weld is slightly lower. This may be due to the fact that the alloy content in the weld is lower than that of the surfacing layer. 

Figure 16 is the surface morphology of the welded joint after corrosion. The corrosion degree of the weld and the base material can be clearly seen in this figure. It shows that the corrosion resistance of the weld is better than that of the base material. However, due to the uneven presence of alloying elements in the weld—more C in the lower part of the weld—the corrosion of the lower part of the weld is greater that that in the upper part. The existence of C forms stable compounds with Cr; thus, more corrosion is found in the lower part of the weld. Corrosion morphology of the weld can be obtained from Figure 16e–h; the figure shows that corrosion increased with proximity to the lower part of the weld.

## 4. Conclusions

This study draws the following conclusions:Using suitable welding parameters, MIG surfacing technology can be used to fabricate a defect-free stainless steel surfacing layer on low carbon steel, and the K-TIG welding method can be applied to join the surface-modified plates. There are no macroscopic defects in the surface-modified plate and its welded joint.The structure of the stainless steel surfacing layer consists of austenite + martensite + ferrite. Lath martensite appeared at the upper part of the weld and flake martensite at the lower end.There are slight differences in the content of alloying elements in the weld. Cr and Ni content was higher at the location close to the surfacing layer, while less-alloying elements were found near the base material at the lower part of the weld. The difference in content of alloying elements affects corrosion resistance, but has little effect on hardness.The corrosion morphology and polarization curve concludes that corrosion resistance of the surfacing layer is the best, the corrosion resistance of the upper zone of the weld is slightly higher than that of the lower part of the weld, and the corrosion resistance of the base material is the worst.

## Figures and Tables

**Figure 1 materials-12-02883-f001:**
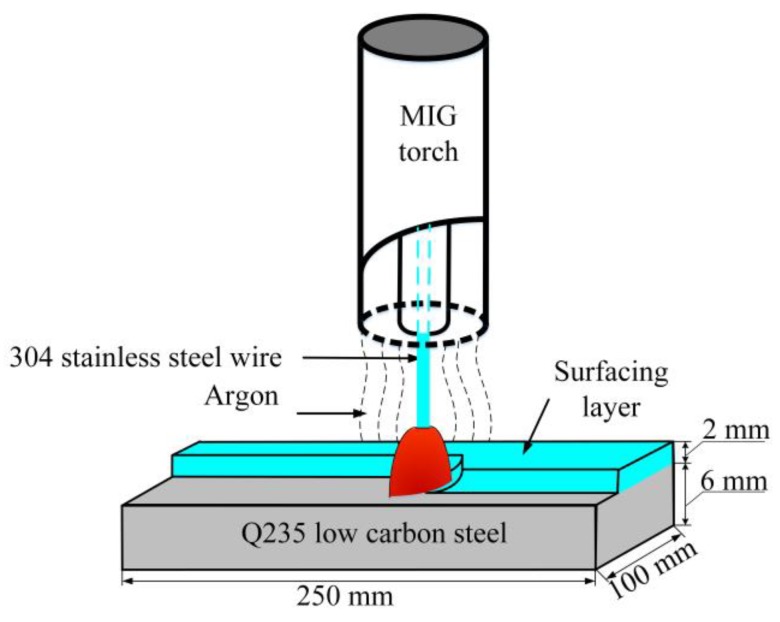
Schematic diagram of the MIG surfacing process.

**Figure 2 materials-12-02883-f002:**
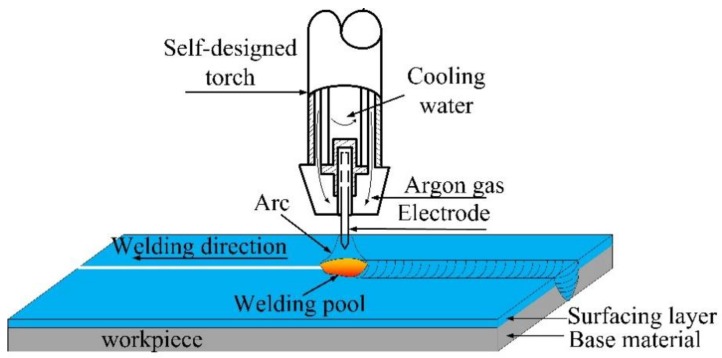
Diagram of the K-TIG welding process.

**Figure 3 materials-12-02883-f003:**
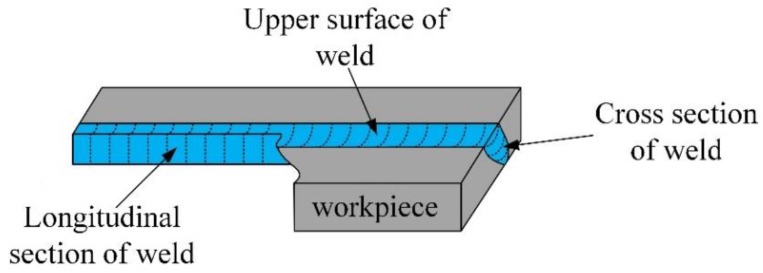
Diagram of each section of the weld.

**Figure 4 materials-12-02883-f004:**
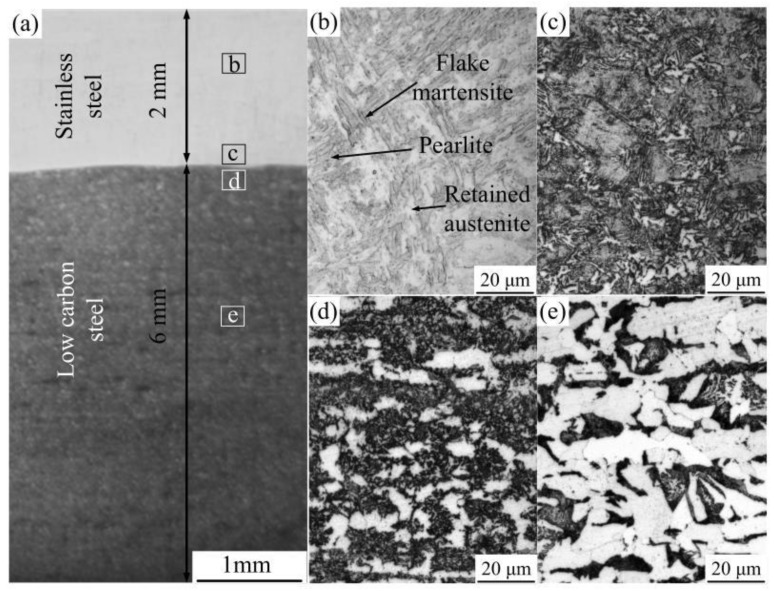
Microstructures of the surface modified plate. (**a**) Macroscopic morphology of the surfacing workpiece; (**b**) microstructure of the stainless steel layer; (**c**) microstructure above the fusion zone; (**d**) microstructure blow the fusion zone; (**e**) microstructure of low carbon steel.

**Figure 5 materials-12-02883-f005:**
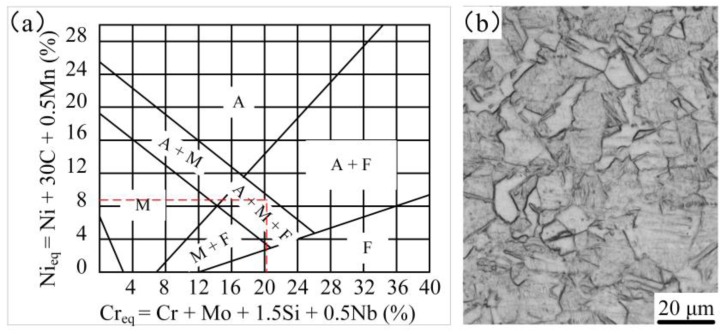
(**a**) Schaeffler diagram, and (**b**) microstructure of the 304 stainless steel.

**Figure 6 materials-12-02883-f006:**
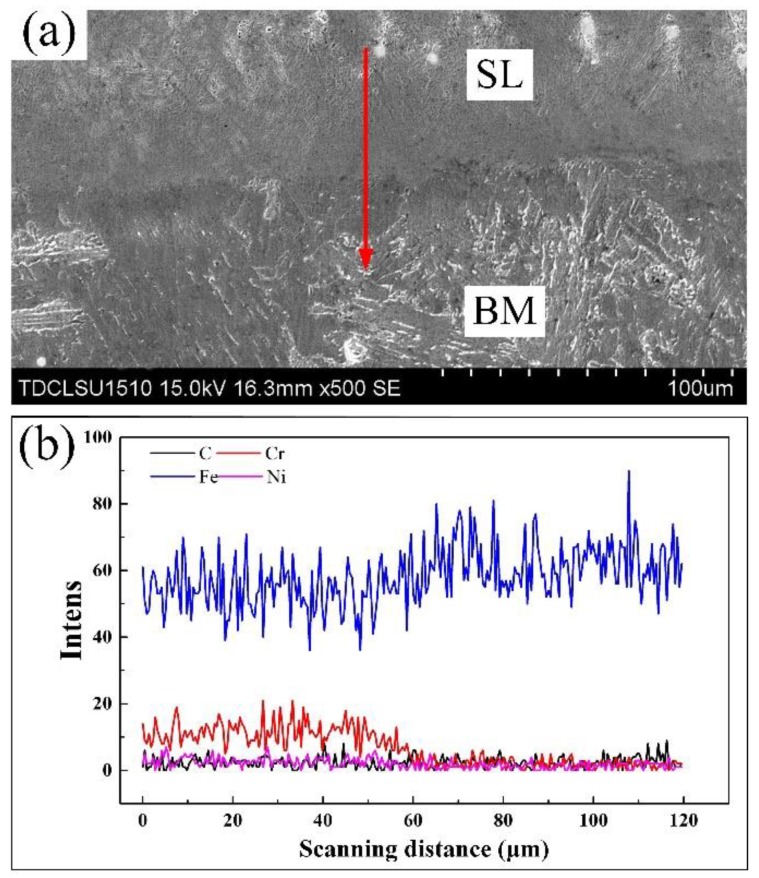
EDS line scan analysis of major alloying elements. (**a**) Scanning position diagram; and (**b**) element distribution.

**Figure 7 materials-12-02883-f007:**
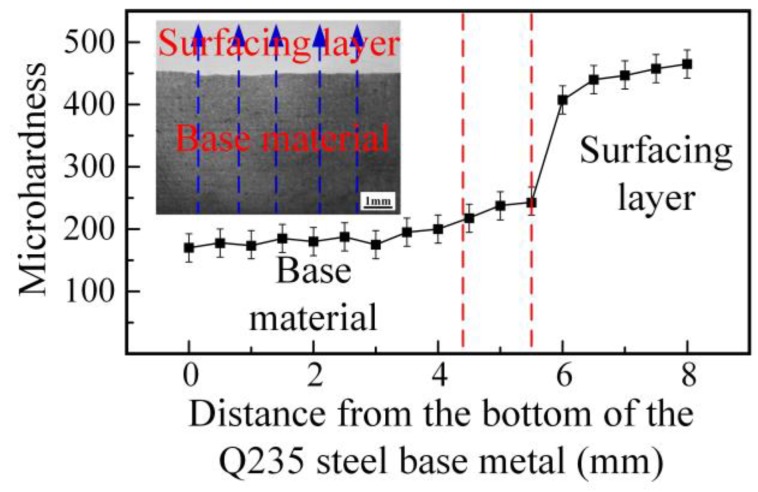
Hardness indentation paths and hardness profile.

**Figure 8 materials-12-02883-f008:**
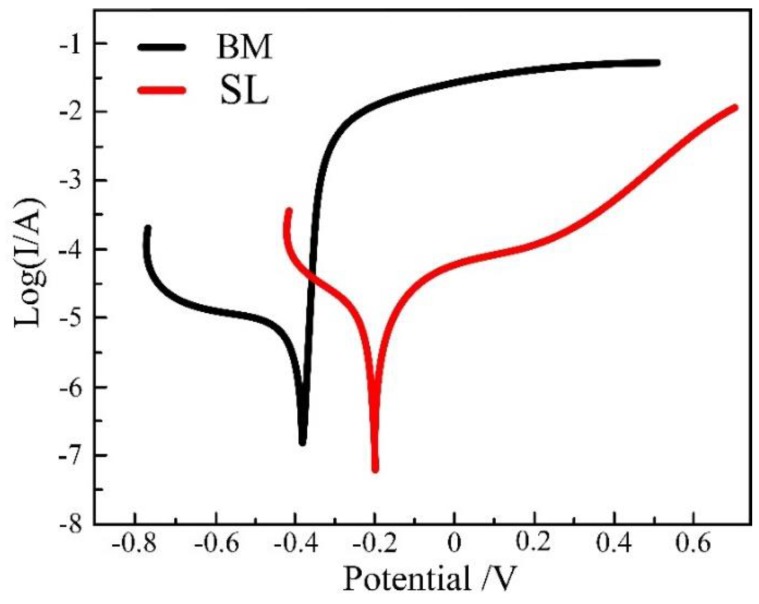
Polarization curve of the surfacing layer.

**Figure 9 materials-12-02883-f009:**
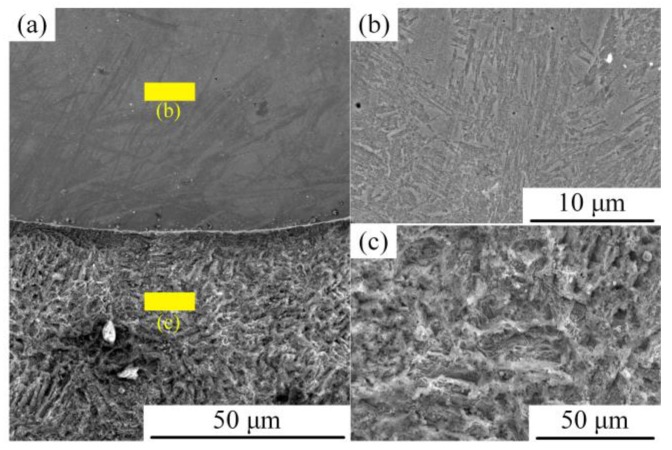
Surface corrosion morphology of modified plate. (**a**) Interface of the surfacing layer and the base material; (**b**) corrosion morphology of stainless steel; and (**c**) corrosion morphology of the base material.

**Figure 10 materials-12-02883-f010:**
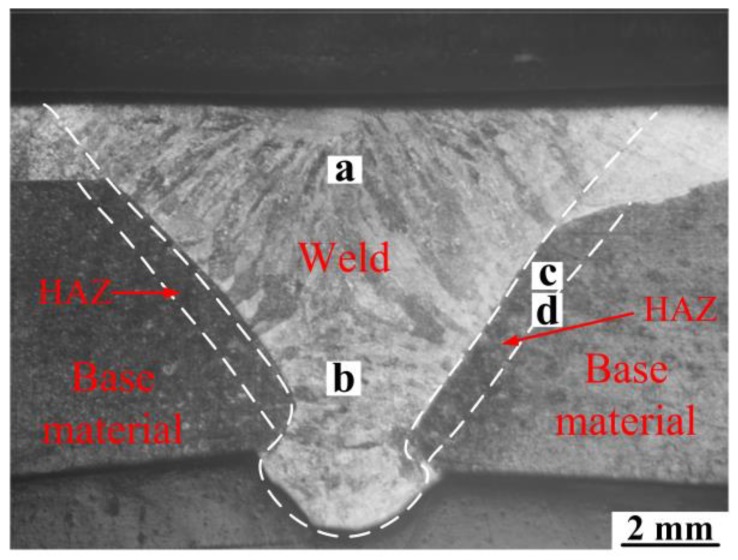
Macroscopic morphology of the welded joint.

**Figure 11 materials-12-02883-f011:**
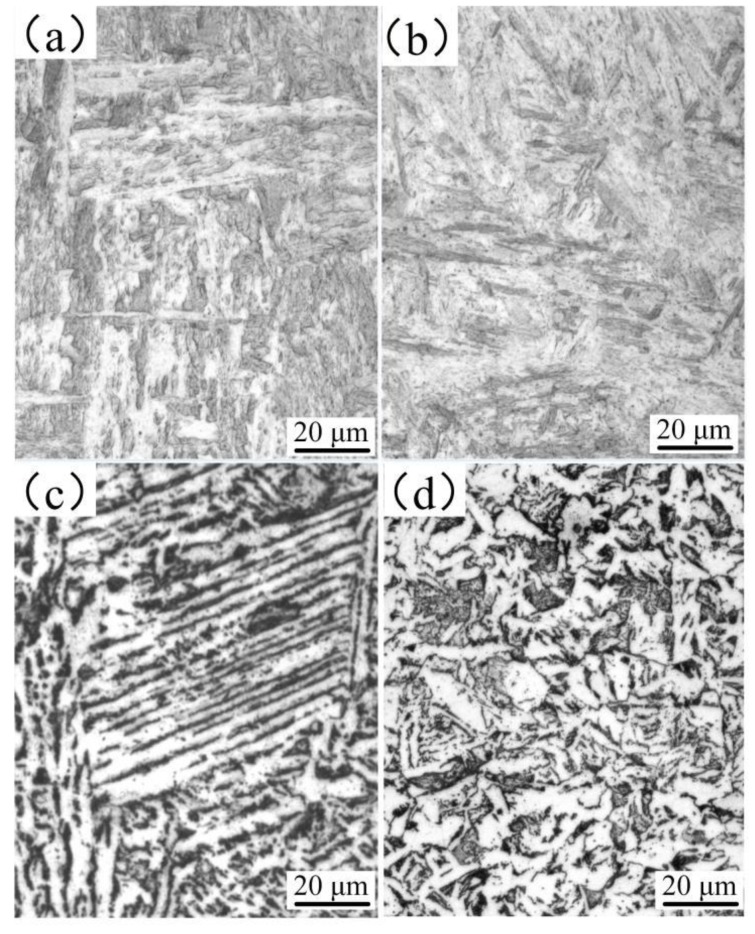
Microstructures of the weld. (**a**) Microstructure of the upper weld; (**b**) microstructure of the lower weld; (**c**) microstructure of the heat-affected zone, and (**d**) microstructure of the base material.

**Figure 12 materials-12-02883-f012:**
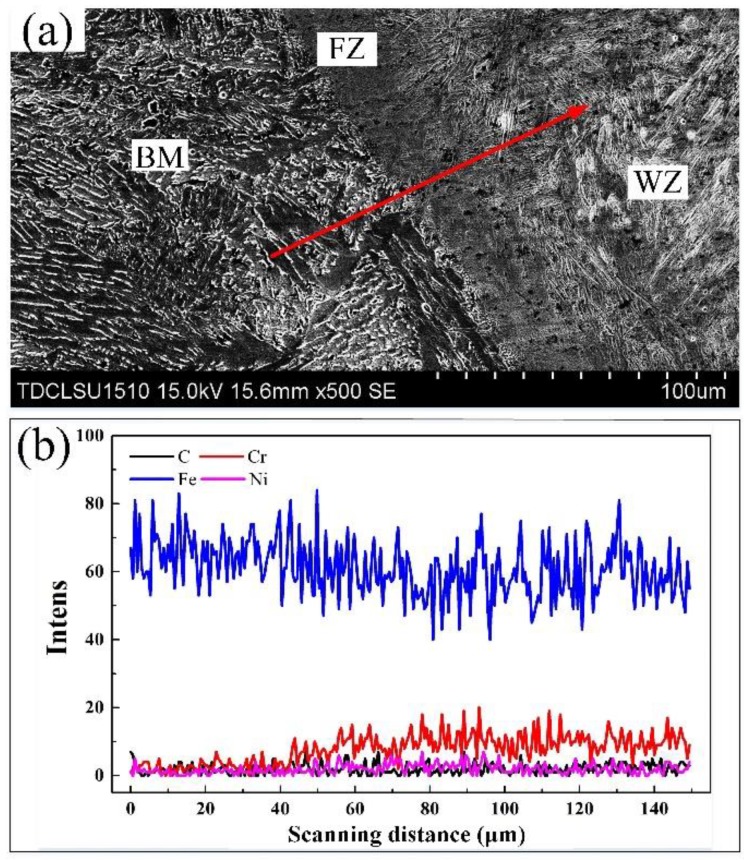
EDS line scan across the weldment. (**a**) Schematic diagram of the weld zone; (**b**) alloying elements’ line scanning.

**Figure 13 materials-12-02883-f013:**
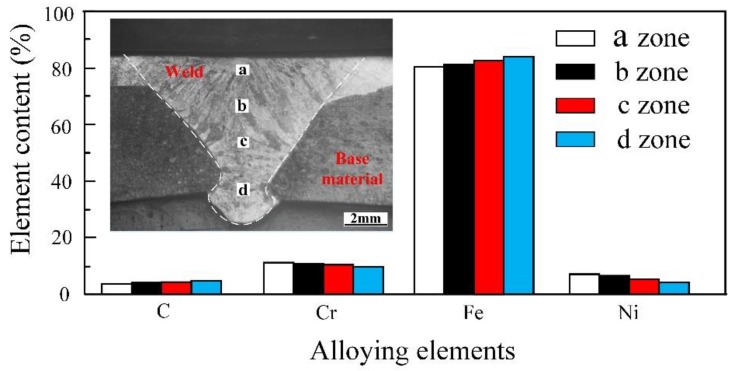
EDS spot scan of the upper weld and lower weld.

**Figure 14 materials-12-02883-f014:**
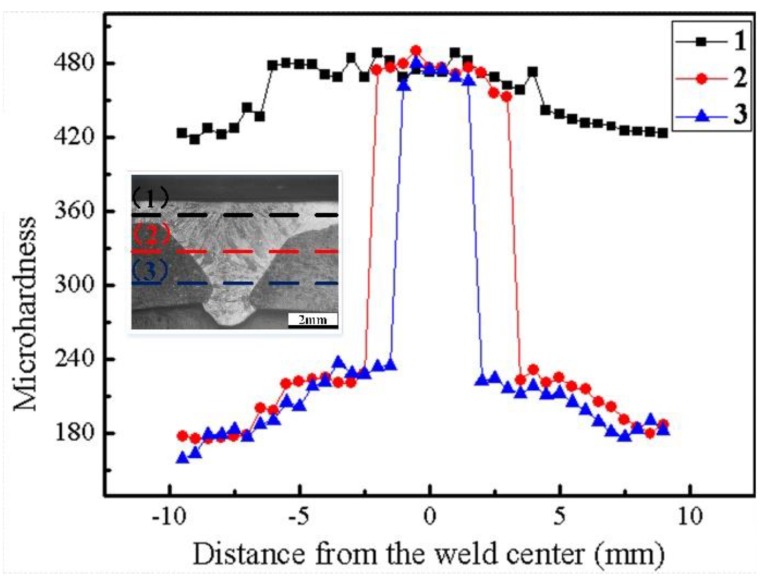
Microhardness distribution across the weld.

**Figure 15 materials-12-02883-f015:**
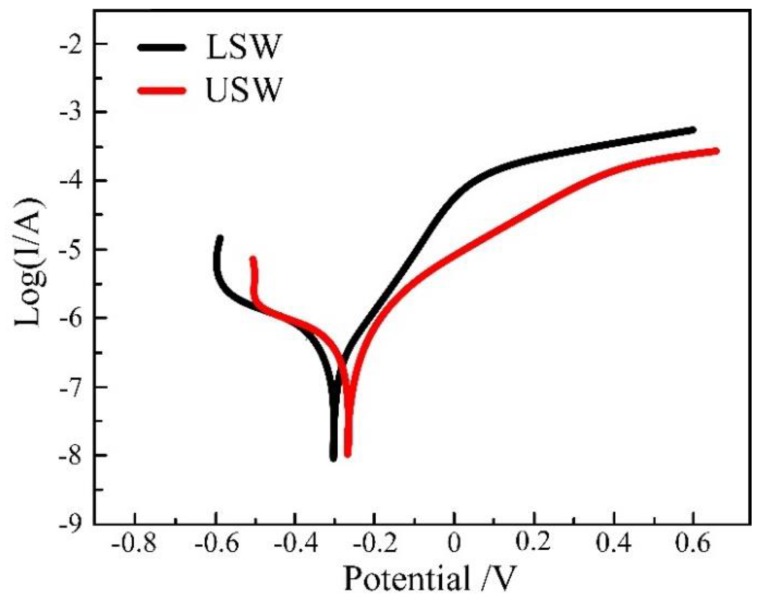
Polarization curve of weld.

**Figure 16 materials-12-02883-f016:**
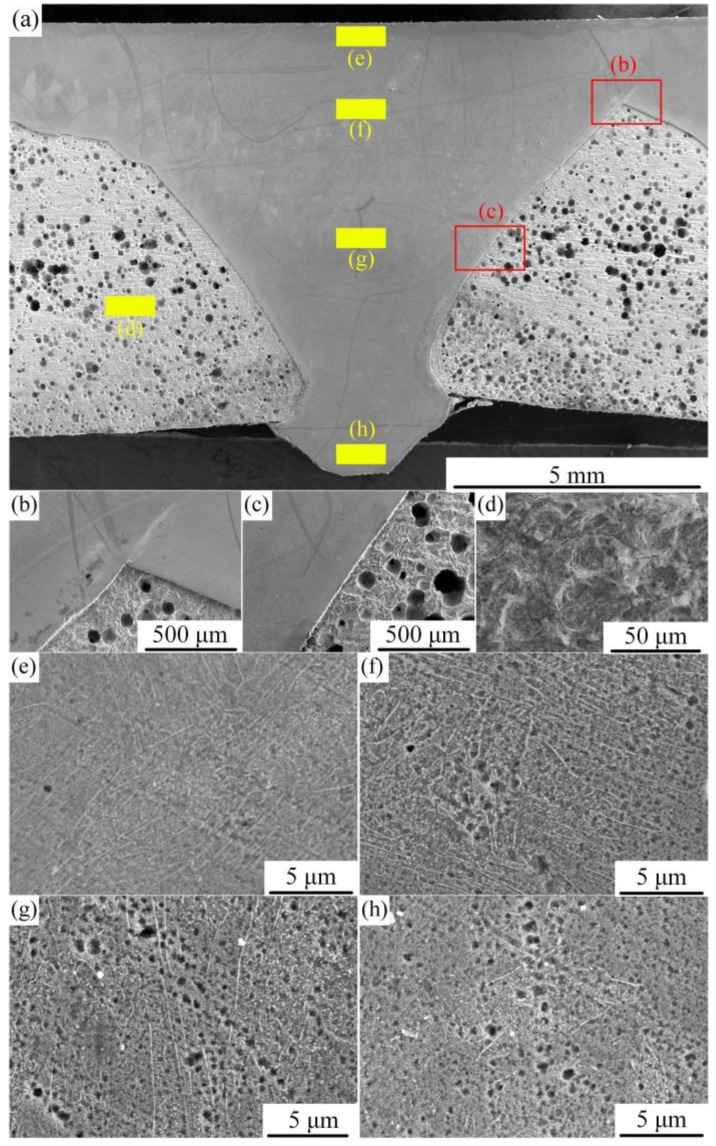
Corrosion morphology of different areas of the weld. (**a**) Macroscopic corrosion morphology of weld; (**b**) corrosion microstructure of zone b; (**c**) corrosion microstructure of zone c; (**d**) corrosion microstructure of zone d; (**e**) corrosion microstructure of zone e; (**f**) corrosion microstructure of zone f; (**g**) corrosion microstructure of zone g; (**h**) Corrosion microstructure of zone h.

**Table 1 materials-12-02883-t001:** The chemical composition of the low carbon steel and stainless steel (wt.%).

Materials	C	Si	Mn	S	P	Al	Cr	Ni	Fe
Q235	0.16	0.3	0.65	0.004	0.01	0.033	–	–	Bal
304	0.05	1.0	1.5	0.03	0.035	–	21	9	Bal

**Table 2 materials-12-02883-t002:** MIG surfacing parameters.

Current (A)	Voltage (V)	Welding Speed (mm/s)	Shielding Gas	Gas Flow (L/min)
150	22	5	Ar	25

**Table 3 materials-12-02883-t003:** Corrosion parameters of low carbon steel, surfacing layer, and fusion zone.

Samples	E_corr_ (V)	I_corr_ (µA/cm^2^)
Low carbon steel	−0.4	−5.762
Surfacing layer	−0.2	−5.122

**Table 4 materials-12-02883-t004:** Corrosion parameters of the longitudinal section and upper surface of the weld.

Samples	E_corr_ (V)	I_corr_ (µA/cm^2^)
Longitudinal section of weld	−0.29	−6.114
Upper surface of weld	−0.23	−5.887

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
