# Peer review of "Microstructure and Properties of Surface-Modified Plates and Their Welded Joints"

_materials, 2019, doi:10.3390/ma12182883_

Round 1

Reviewer 1 Report

The paper deals with the fabrication of stainless steel surfacing layer on low carbon steel by MIG method, aiming to enhance the surface properties of the low carbon steel. Subsequently, K-TIG welding was adopted to weld surfaced modified plates. Mechanical and corrosion properties were tested along with a study of the corresponding microstructures. The paper is well structured and may be accepted for publication after minor correction

The minute details such as " Metallographic samples of the cross sections of the surface modified plates and its welded joints were ground using 80#, 120#, 240#, 400#, 800#, 1200#, 1500#, and finally 2000# abrasive papers " may not be required in experimental section (2.3. Microstructure characterization). The caption of the figure 4 is not clear. There are five images in the figure. However, all are not accounted in the figure caption. “Linear sweep voltammetry technique” is said to be used by the authors (line 124). However, potentiodynamic polarization test results are only shown in the manuscript. Please correct this discrepancy. Have the authors conducted any weight measurements prior and after the acid immersion tests that they have conducted?

Reviewer 2 Report

The paper is presented very well. However, some of the important details were left out and including these will help understand and utilise the results better.

English language corrections and additional information where required are marked in the paper. Please add them.
